# A Novel Dimeric Conotoxin, FrXXA, from the Vermivorous Cone Snail *Conus fergusoni*, of the Eastern Pacific, Inhibits Nicotinic Acetylcholine Receptors

**DOI:** 10.3390/toxins14080510

**Published:** 2022-07-26

**Authors:** Ximena C. Rodriguez-Ruiz, Manuel B. Aguilar, Mónica A. Ortíz-Arellano, Helena Safavi-Hemami, Estuardo López-Vera

**Affiliations:** 1Posgrado en Ciencias del Mar y Limnología, Universidad Nacional Autónoma de México, Ciudad de México 04510, Mexico; himena_dark@hotmail.com; 2Laboratorio de Neurofarmacología Marina, Departamento de Neurobiología Celular y Molecular, Instituto de Neurobiología, Universidad Nacional Autónoma de México, Querétaro 76230, Mexico; maguilar@unam.mx; 3Laboratorio de Malacología, Facultad de Ciencias del Mar, Universidad Autónoma de Sinaloa, Mazatlán 82000, Mexico; manabel@uas.edu.mx; 4Department of Biomedical Science, University of Copenhagen, 1172 Copenhagen, Denmark; safavihelena@sund.ku.dk; 5Department of Biochemistry, University of Utah, Salt Lake City, UT 84112, USA; 6Laboratorio de Toxinología Marina, Unidad Académica de Ecología y Biodiversidad Acuática, Instituto de Ciencias del Mar y Limnología, Universidad Nacional Autónoma de México, Ciudad de México 04510, Mexico

**Keywords:** alpha d-conotoxin, dimeric-peptides, nAChR, *Conus* snails, *Conus fergusoni*

## Abstract

We isolated a new dimeric conotoxin with inhibitory activity against neuronal nicotinic acetylcholine receptors. Edman degradation and transcriptomic studies indicate a homodimeric conotoxin composed by two chains of 47 amino acid in length. It has the cysteine framework XX and 10 disulfide bonds. According to conotoxin nomenclature, it has been named as αD-FrXXA. The αD-FrXXA conotoxin inhibited the ACh-induced response on nAChR with a IC_50_ of 125 nM on hα7, 282 nM on hα3β2, 607 nM on α4β2, 351 nM on mouse adult muscle, and 447 nM on mouse fetal muscle. This is first toxin characterized from *C. fergusoni* and, at the same time, the second αD-conotoxin characterized from a species of the Eastern Pacific.

## 1. Introduction

*Conus* is a genus of carnivore marine snails that has more than 800 species, living mostly in the Indo-Pacific Ocean; in Mexico about 70 species living in the coasts of Pacific Ocean and Gulf of Mexico have been registered [1,2,3]. As they are slow moving animals, they have developed, as an evolutionary strategy, the use of venom for defense against their predators and as a weapon to capture prey, which involves worms, mollusks, and fishes. To administrate the venom, they have an innovative structure, the radula teeth modified as harpoon-like hypodermic needles that allow them to inject the venom that is synthetized in the venom duct [4,5].

The venom from each *Conus* species could have more than 100 active components that are known as conotoxins. Most of them are small peptides composed of 10 to 40 amino acids, which makes their characterization and synthesis relatively easy. In addition, they are stable peptides due to disulfide bonds, resulting from different cysteine frameworks [6,7]. Conotoxins have been classified into genetic superfamilies according to the signal peptide of their precursors and in pharmacological families for every molecular target on which they act [8]. The great affinity towards their molecular target allows them to discriminate between the different subunits that form distinct subtypes of membrane receptors [9].

Alpha-conotoxins can modulate the muscular and neuronal nicotinic acetylcholine receptors (nAChRs). These are one of the most studied families of conotoxins because they can distinguish between the numerous subtypes of nAChRs, expressed along the central and peripheral nervous systems. These receptors have different pharmacological and electrophysiological properties; there is evidence that nAChRs are involved in pain processes and human diseases such as Alzheimer’s, Parkinson’s, schizophrenia, and autism [8,10,11]. These features make α-conotoxins suitable candidates for therapeutic agents for human diseases. Each *Conus* snail has at least one α-conotoxin in its venom. There are more than 64 conotoxins belonging to several gene superfamilies that are reported to act as antagonist for the different nAChR subtypes and that are classified as: α-, αA-, ψ-, αS-, αB-, αC-, and αD-conotoxins [12,13,14].

α-Conotoxins belonging to the D superfamily were first described by Loughnan et al. [15]. They are large conotoxins whose main characteristic is to form dimers with varying levels of sequence and posttranslational heterogeneity. Each chain has almost 50 amino acid residues, including 10 cysteines that form disulfide bonds [15,16]. To the present, 25 αD-conotoxins have been reported from vermivorous species found in the Indo-Pacific [17]. However, in 2019 the first αD-conotoxin from a species that inhabits the Mexican Pacific Coast in the Eastern Pacific, *C. princeps*, was reported [18].

In this work, we report the identification, characterization, and activity of a novel conotoxin, αD-FrXXA, isolated from the venom of the vermivorous cone snail *C. fergusoni*, collected off the Coast of Mazatlán, Sinaloa, México. Peptide αD-FrXXA has a slightly inhibitory activity preference against neuronal rather than muscular nAChRs. The biochemical characterization indicates that it belongs to the αD-superfamily. Conotoxins with activity against nAChRs can act as new molecular tools for cellular and molecular studies in mammals’ nervous systems in addition to their pharmacological potential use. This work provides more evidence that this type of dimeric conotoxins are synthesized not only by vermivorous species of the Indo-Pacific, but also by several vermivorous species of the Eastern Pacific, as has been shown for *C. princeps*. Finally, it opens the opportunity to discover more dimeric toxins in species inhabiting Mexican coasts, because it seems that αD-conotoxins could be abundant in *Conus* venoms from Eastern Pacific species.

## 2. Results

### 2.1. Fractionation of C. Fergusoni Venom

Fractionation of *C. fergusoni* venom by RP-HPLC gave 52 fractions. All of them were tested in the electrophysiology assay and four fractions had inhibitory activity over 90% against α7 nAChRs (marked with an * in Figure 1A). The fourth fraction was the most active and was chosen for further experiments. After sub-fractionation by RP-HPLC a double signal was detected and manually collected in two 1.5 mL Eppendorf tubes (Figure 1B).

### 2.2. Molecular Mass and Amino Acid Sequence Analysis

The mass spectra of the two subfractions obtained from the fourth active fraction (without reduction and alkylation) are shown in Figure 2 and named 4a and 4b. The spectrum of subfraction 4a displays minor (z = +1) and major *m*/*z* (z = +2) signals 11,074.174 and 5513.533 (Figure 2A), that correspond to average masses of 11,073.164 Da and 11,025.046 Da, respectively; these masses differ by 48.118 Da, which might correspond to a γ-carboxylation. For subfraction 4b, the minor (z = +1) and major *m*/*z* (z = +2) average signals of 11,115.118 and 5481.605 correspond to average masses of 11,114.108 Da and 10,961.190, respectively (Figure 2B); in this case, the difference (152.918 Da) is more difficult to explain and will be dealt with in the Discussion section.

A sequence of 26 amino acids was obtained by Edman degradation from component 4b (the most abundant component from subfractionation (Figure 1B)). Positions 6, 17, 18 and 23 were blank cycles, which we assume correspond to cysteine residues. Position 8 and 9 presented another signal which may correspond to amino acid variation S8P and T9R (Figure 3).

Therefore, we decided to search for sequences like that obtained from 4b in the transcriptome of the venom duct of *C. fergusoni* (to be published). The transcriptomic study allowed us to identify three precursors that contain sequences similar at the mature toxin region compared to the amino acid sequence yielded by Edman degradation of 4b. All precursor sequences are 92 amino acids in length and include the signal sequence, a pro region, and a mature toxin (Figure 4). Precursors 1 and 2 are very similar and contain the same predicted mature toxin; therefore, from now on, we will only refer to toxin 1 (from precursors 1 and 2) and toxin 2 (from precursor 3).

### 2.3. Similarity Search

BLASTp similarity search for sequence 1 gave 26 BLAST hits with an E value over 1 × 10^−20^. There was an identity percentage of 69.57%, an E value of 3 × 10^−29^ and a query coverage of 100% for conotoxin precursor superfamily D from *C. ermineus*. In addition, it had an identity of 65.59%, an E value of 2 × 10 ^−37^ and 100% of query coverage for Cp20.4 from *C. capitaneus*, and an identity of 64.13% with an E value of 1 × 10 ^−36^ and query coverage of 100% for VxXXA from *C. vexillium* (data not shown).

Results for sequence 2 gave 25 BLAST hits with an E value over 1 × 10^−20^. The identity percentage of 72.83% showed an E value of 5 × 10 ^−30^ and a query coverage of 100% for conotoxin precursor superfamily D from *C. ermineus*. It also had an identity of 67.74% an E value of 1 × 10^−38^ and 100% query coverage for Cp20.4 from *C. capitaneus*, and an identity 61.29% with an E value of 7 × 10^−36^ and 100% query coverage for GeXXA from *C. generalis* (data not shown).

### 2.4. Electrophysiology Assay

Component 4b, which was the most abundant, was tested as a first approximation on hα7 nAChR expressed in *Xenopus* oocytes at 1 μM. Component 4b showed an 100% inhibitory activity on the currents elicited by ACh. Hence, we tested component 4b on several subtypes of nAChRs (hα4β2, hα3β2, m(α1)_2_β1δγ, and m(α1)_2_β1δε). Toxin concentration-response curves were generated for all the nAChRs subtypes. Notably, the IC_50_ range was in the order of nM for all subtypes of nAChRs, with a slightly higher affinity for α7 nAChR (Figure 5). In order to demonstrate distinct dissociation rates on the different subtypes of nAChR, we evaluated 600 nM concentration of component 4b (from now on referred to as αD-FrXXA) which is close to the highest IC_50_ values. αD-FrXXA demonstrated different dissociation rates; the slower dissociation was for hα7 subtype (almost irreversible) meanwhile the fastest was for hα4β2 subtype (Figure 6).

## 3. Discussion

In this work, we isolated and characterized a new member of the αD-conotoxins from the venom of the vermivorous marine snail *C. fergusoni*, which inhabits the Mexican Pacific Coast. Within the venom of *C. fergusoni*, four signals with inhibitory activity were found. However, the present study focused on characterizing the fourth signal due to its high inhibitory activity on nAChR, its abundance and apparent purity. The fact that one signal with two peaks was obtained during the repurification process, indicated a compound that due to the similarity of its components, would be difficult to separate by means of polarity difference between them.

To know more about this compound, left and right components of αD-FrXXA were collected to obtain their molecular weight prior performing more experiments. It should be mentioned the most abundant signals in the mass spectrum correspond to the charge +2 species, whereas the less abundant species have charge +1. The deconvoluted values for both samples were: 11,025.046 Da and 11,073.164 Da for component 4a and 10,961.190 Da and 11,114.108 Da for 4b (Figure 2). These molecular masses indicate it was a conotoxin with a high molecular weight in relation to the common α-conotoxins from the A-superfamily.

Transcriptomics analysis indicated that in the venom of *C. fergusoni* there are two different sequences (derived from three distinct precursors) which matched in part the one obtained by Edman degradation. This shows us two mature toxins with 47 amino acids in length and differing from each other only by six amino acids. Sequence 1 has a theoretical average mass of 5297.18 Da, while Sequence 2 has 5435.30 Da (average), according to Ion Source Peptide Mass Calculator [19], considering 10 cysteines oxidized and free C- and N-termini.

Since only Tyr residues were observed at position 26 during Edman sequencing, we focused on sequence 2, which contains Tyr at that position; we calculated the molecular average mass for the homodimer formed by sequence 2 and the variation of two amino acids of these dimers detected by Edman sequencing. Then, we compared them with the experimental masses obtained. As a result, we found that the compound with experimental mass of 11,025.046 Da might be formed by a homodimer of the sequence 2 with one TFA adduct and one potassium adduct, resulting in a molecular mass of 11,023.70 Da; also, it could be formed by a homodimer of sequence 2 with the variation of P8S and R9T, with two TFA adducts, one potassium adduct and one oxygen and a resulting molecular mass of 11,022.47 Da.

The experimental mass of 11,073.164 Da might be formed by a homodimer of the sequence 2, with one TFA adduct, one potassium adduct, one sodium adduct and two oxygen resulting in a calculated mass of 11,077.68 Da. The compound with an experimental mass of 10,961.190 Da might be formed by a homodimer of the sequence 2 with the variation of P8S and R9T, with two TFA adducts, resulting in a calculated mass of 10,968.38 Da. Finally, the compound with experimental mass of 11,114.108 Da might be a homodimer of sequence 2, with two TFA adducts and two oxygens resulting in a calculated mass of 11,114.62 Da. These combinations gave a difference of +1.35 Da, +2.58 Da, −4.52 Da, −7.19 Da, and −0.51 Da, respectively, when comparing the experimental masses and the calculated masses of the formed dimer with these modifications (Table 1). The difference of the five values falls within the error range of 0.1% for MALDI-TOF for peptides of this length.

The formation of adducts, either with solvent molecules, alkali ions or other metals, or with other contaminating components, is frequently observed in the mass spectra of both ESI and MALDI analysis [20]. The occurrence of cationic and anionic adducts in MALDI spectra varies according to the number and polarity of the charged groups of the analyte [21].

Particularly, most ions observed in modern mass spectrometry are adduct ions or pseudomolecular ions such as [M + H]^+^, [M + NH_4_]^+^, [M + Na]^+^ or [M + K]^+^ [20]; Na^+^ adducts are one of the most common in biological samples [22]. In addition, TFA adducts are reported as part of the repeating units commonly observed in background interferences [20]; these adducts have already been reported by Loughnan et al., and Hernández-Sámano et al., during the analysis of the αD-conotoxins from *C. vexillum* and *C. princeps*, respectively [15,18].

When performing the alignment in BLASTp, it indicated the two sequences correspond to the superfamily D of conotoxins. This family was first reported by Loughnan [15] and the main characteristic of the αD-conotoxins is that they are formed by dimers which can be hetero-, homo- or pseudo-homodimers (which vary only by post-translational modifications) [15]. The αD superfamily also included a conserved motif WGRCC, the same motif that can be seen in the amino acids 14–18. Based on that evidence, we decided to use the name αD-FrXXA following the proposed nomenclature for conotoxins [23].

It is known most conotoxins present post-translational modifications. The three αD-conotoxins from *C. vexillium*: VxXXA, VxXXB, and VxXXC have proline hydroxylation and gamma glutamate carboxylation; on the other hand, PiXXA isolated from *C. princeps* shows the hydroxylation of proline [15,18]. This makes hydroxylation a common post-translational modification in αD-conotoxins and it could be that αD-FrXXA is not the exception; it might have oxygens according to the calculated values from the molecular masses mentioned above (fourth paragraph).

Regarding how the dimer is formed, it can be seen in Figure 2 that the most abundant signals are the ones whose experimental weight had +2 charges and, in both cases, correspond to the presence of a homodimer formed by Sequence 2. However, there were variations in the amount of adducts; this indicates αD-FrXXA is a homodimer formed by two chains of Sequence 2.

To a lesser extent, a homodimer was also discovered that might be formed by two chains of Sequence 2 with the variation of amino acids that was found by Edman sequencing (P8S and R9T). This could be due to intraspecific variations in the venom, since the transcriptome was obtained from the venom duct of a single organism, but the venom used for the biochemical analysis and electrophysiological assays was the result of a mixture of the venom from different organisms.

The fact the same species has several amino acid sequences from the same conotoxins superfamily is well known. In the case of the D-superfamily it was found that *C. capitaneus, C. miles, C. mustelinus*, and *C. vexillium* have at least three different monomers reported in ConoServer. However, their dimer conformations were experimentally demonstrated only for *C. vexillium* [15], *C. capitaneus* [12], *C. generalis* [24], and *C. princeps* [18]. Therefore, despite having two monomers in the *C. fergusoni* venom transcriptome, these chains could be coupled in different ways and there could be five different dimers.

From the marine snail species that have αD-conotoxins, only the pharmacological target of three species have been identified: *C. vexillium* with VxXXA, VxXXB and VxXXC; *C. generalis* with GeXXA, and *C. princeps* with PiXXA. All of them have inhibitory activity against nAChRs, being the neuronal subtypes those with the highest affinity (Table 2) [15,18,24]. Although *C. mustelinus* and *C. capitaneus* have also shown inhibitory activity against nAChRs, the affinity of their toxins has not been studied yet [16].

According to the results obtained, dimer αD-FrXXA has the higher affinity for the neuronal α7 nAChR subtype with an IC_50_ of 125 nM. However, blockade was observed with an IC_50_ lower than 500 nM for α3β2 and the muscular subtypes.

It should be mentioned that for α7 and α3β2 nAChR subtypes, blockage presented by the dimer was irreversible since, despite the passage of time and washing, the amplitude of the current does not increase (Figure 6). On the other hand, in muscular subtypes (adult and fetal) the dimer was slightly reversible although the current amplitude could not return to its initial values, which indicates that the toxin was not completely separated from the receptor (Figure 6). Finally, for the α4β2 subtype the dimer was reversible almost entirely within a few minutes of washing which indicates the toxin has less affinity to this receptor. (Figure 6). Reversibility in αD-conotoxins has been observed in *C. vexillum* venom where inhibition on α7 and α3β2 nAChR subtypes was reversible [15]. It is important to mention that αD-conotoxins as well as Ψ-conotoxins are nAChR blockers and that both families have the characteristic to not be competitive antagonists. In other words these toxins do not bind to the orthosteric agonist binding site as happens with other α-conotoxins (α, αA, αS, and αC) [12,24].

Yang et al. mentioned αD-conotoxins, in particular GeXXA, that binds to the outer surface of the receptor more than to the intermembrane domain, as a “lid-covering”nAChR inhibitor [25]. This blockade not only seals the pore entrance and prevents the passage of cations, but it also prevents the conformational change caused by the opening of the channel when the agonist binds the channel. This type of activity may explain why the sigmoid slope observed is so steep. In the case of GeXXA, it has been observed that the binding has preference towards the β subunits rather than the α [25].

In the case of αD-conotoxins tested on human receptors, GeXXA has affinity for the α9α10 subtype with an IC_50_ of 28 nM while for α7 subtype this is 210 nM. For αD-FrXXA a similar value was obtained for α7 (125 nM) [25]. In the case of PiXXA, it has activity over nAChR α7 but with a micromolar concentration which is greater than that presented for αD-FrXXA [18].

It is worth mentioning that αD-FrXXA may have a higher affinity for nAChR from rat specimens with picomolar values; this is the case for VxXXB on rα7 subtype. There is similar evidence for GeXXA, whose activity over rα9α10 is ten times higher compared to the human subtype [26].

## 4. Conclusions

In summary, αD-FrXXA adds evidence that these dimeric toxins may play a biological role in the survival of species that inhabit the Eastern Pacific Ocean. Recently, the first αD-conotoxin isolated from a species inhabiting the Eastern Pacific was described from *C. princeps*, by Hernández-Sámano et al. [18].

## 5. Materials and Methods

### 5.1. Isolation of Crude Venom Extract from C. fergusoni

*C. fergusoni* specimens were collected in soft substratum off the coasts of Mazatlán, Sinaloa, México and frozen at −70 °C. Five venom ducts were dissected using fine tweezers and homogenized in 10 mL of 40% (*v/v*) acetonitrile (ACN) and 2% (*v*/*v*) trifluoroacetic acid (TFA) with a tissue homogenizer (Biospec Products, Tissue Tearor, Model 985370). The resulting mix was centrifuged at 14,000× *g* for 10 min. The supernatant was collected by decantation and the protein content was quantified with a Nanodrop spectrophotometer at 280 nm wavelength (Implen NanoPhotometer NP80). The venom extract was lyophilized and stored until its use for fractionation.

### 5.2. Venom Extract Fractionation by RP-HPLC

Aliquots of 8 mg of crude venom were dissolved with 1 mL of Solution A (aqueous solution with 0.1% (*v/v*) TFA) before fractionation by Reversed-Phase High Performance Liquid Chromatography (RP-HPLC) (Infinity 1260 Series Agilent Technologies HPLC System; Santa Clara, CA, USA) using a Vydac Reverse-Phase Peptide and Protein C_18_ column (218TP54, 5 µm particle size, 4.6 mm, 250 mm) attached to a Vydac Reverse-Phase Peptide and Protein C_18_ Precolumn (218GK54, 5 µm particle size, 4.6 mm, 10 mm).

Fractions of the crude venom extract were eluted with an isocratic step of 5% Solution B (90% ACN in water containing 0.085% *(v/v)* TFA) for 5 min, followed by a linear gradient from 5% to 100% (*v/v*) of Solution B in 95 min, at a flow rate of 1 mL/min. Further purification of αD-FrXXA was performed by RP-HPLC using an isocratic step at 17% of Solution B for 5 min, followed by a linear gradient from 17% to 37% (*v/v*) of Solution B, over 65 min at a flow rate of 1.0 mL/min. The elution profiles were monitored at a wavelength of 220 nm. Each signal was collected manually.

### 5.3. Quantification of αD-FrXXA

The quantification of αD-FrXXA was first performed, by relating the area under the curve obtained by RP-HPLC for 5 nmol of the synthetic conotoxin α-RgIA (16,829,353 units) with respect to the area obtained for αD-FrXXA. The samples sent to obtain molecular mass and sequence were calculated in this way.

When the molecular weight of αD-FrXXA was already available, the weight in milligrams was used to have a more precise value of the total nmol to perform the electrophysiological assays. This was done assuming that 5 nmol of α-RgIA with an area under the curve of 16,829,353 units is equivalent to 7.8 μg. With this value, we calculated the amount of µg for αD-FrXXA in relation to its area under the curve; subsequently, this value was divided by the weight (kDa) obtained by the mass spectra.

### 5.4. Characterization of αD-FrXXA Conotoxin

Molecular mass and amino acid sequence determination.

Two pmol of the active peptide were analyzed by matrix-assisted laser desorption ionization technique (MALDI-TOF; CDMX, Mexico) with an α-cyano-4-hydroxycinnamic acid matrix at Chemistry Institute, UNAM, facilities.

For amino acid sequence, one nmol of αD-FrXXA was analyzed by automated Edman degradation using a PPSQ-31A Protein Sequencer (Shimadzu Scientific Instruments; Tokio, Japan) by Dr. Fernando Zamudio at Biotechnology Institute, UNAM. Transcriptomic studies will be published.

### 5.5. Data Base Analysis

The Protein BLAST program was used to search the non-redundant protein sequences with organism *Conus* (taxid: 6490), and the algorithm BlastP (protein-protein BLAST) with the default settings for the rest of the parameters.

### 5.6. Propagation of the Genetic Material for nAChRs Subunits

cDNA clones for neuronal human α3, α4, α7 and β2, and muscular mouse α1, β1, δ, γ, ε nAChR subunits were kindly provided by Dr. Michael McIntosh (University of Utah, Salt Lake City, UT, USA).

The plasmids of the nAChRs subunits were inserted in XL-1 Blue competent cells and were purified by kit Wizard Plus SV Minipreps DNA Purification System (Promega; Madison, WI, USA). Only the cDNA of neuronal nAChRs subtypes was linearized using Not I enzyme and purified with EZ-10 Spin Column PCR Purification Kit (Bio Basic; CDMX, Mexico). Transcription was done following the T7 mMessage mMachine Kit (Applied Biosystem; Foster City, CA, USA) protocol and purified with the RNeasy Mini Kit (QIAGEN; Redwood City, CA, USA).

### 5.7. Electrophysiology Characterization on nAChRs

#### 5.7.1. Oocytes Isolation

*Xenopus laevis* frogs were anesthetized with 2% tricaine methanesulfonate (MS-222) for 30 min before performing microsurgery. Oocytes were placed in a 50 mL-Falcon tube with OR-2 solution (82.5 mM NaCl, 2.5 mM KCl, 1.0 mM MgCl_2_·6H_2_O and 5 mM HEPES, pH 7.5 adjusted with 10 N NaOH) and washed at least 5 times with 40 mL to eliminate impurities.

To defolliculate the oocytes, 0.75 mg/mL of *Clostridium histolyticum* Collagenase A (Roche, 10103586001) were added into 20 mL OR-2 solution and stirred for half an hour at room temperature. Then, the oocytes were washed at least 5 times with 40 mL of OR-2 solution to eliminate Collagenase A. Stage VI oocytes were manually selected and kept in ND96 extracellular solution (96.0 mM NaCl, 2.0 mM KCl, 1.0 mM MgCl_2_·6 H_2_O, 1.8 mM CaCl_2_·2 H_2_O, 5 mM HEPES, pH 7.3) added with Penicillin/Streptomycin (100 U/100 µg)/mL and 100 µg/mL Gentamicin at 15 °C.

#### 5.7.2. Heterologous Expression

One day after harvesting, 5.2 ng and 2.5 ng of cDNA for the adult and fetal muscle nAChR subtypes, respectively, were injected into oocytes with a Nanoliter 2000 (World Precision Instruments; Guadalajara, Mexico). For the neuronal nAChR subtypes α7, α3β2 and α4β2, 32.2, 32.2, and 18 ng (total) of RNA were injected, respectively.

#### 5.7.3. Electrophysiological Assay

Two-electrode voltage clamp technique was used for the recordings of neuronal and muscular nAChRS expression 48 h upon injection. Glass microelectrodes (borosilicate capillaries) were filled with 3 M KCl. Injected oocytes were placed in a 30 µL recording chamber and membrane potential was clamped at −70 mV. ND96 solution was perfused by gravity. Currents were generated applying pulses of 1 s of acetylcholine (ACh) per min. Concentration of ACh was 1 µM for the fetal muscle subtype, 10 µM for the adult muscle, 200 µM for neuronal α7, and 100 µM for α3β2 and α4β2 neuronal subtypes.

To test the toxin, the perfusion of ND96 was stopped for 5 min and different concentrations of αD-FrXXA were directly pipetted into the recording chamber. After 5 min of interaction, the toxin-receptors, perfusion and pulses were started over. Current amplitude was measured before and after toxin administration. The difference between them was converted to an inhibition percentage, taking the amplitude before toxin application as 100%.

Data acquisition was done with the program LabView (National Instruments; Ciudad Juarez, Mexico). An average of three oocytes was used to obtain the percentage for each data. Concentration-response curves were fit to the equation % response + 100[1 + (toxin concentration/IC_50_) n_H_], where n_H_ is the Hill coefficient, and the graphics were obtained with GraphPadPrism Software, Version 9.1.0, San Diego, CA, USA.

## Figures and Tables

**Figure 1 toxins-14-00510-f001:**
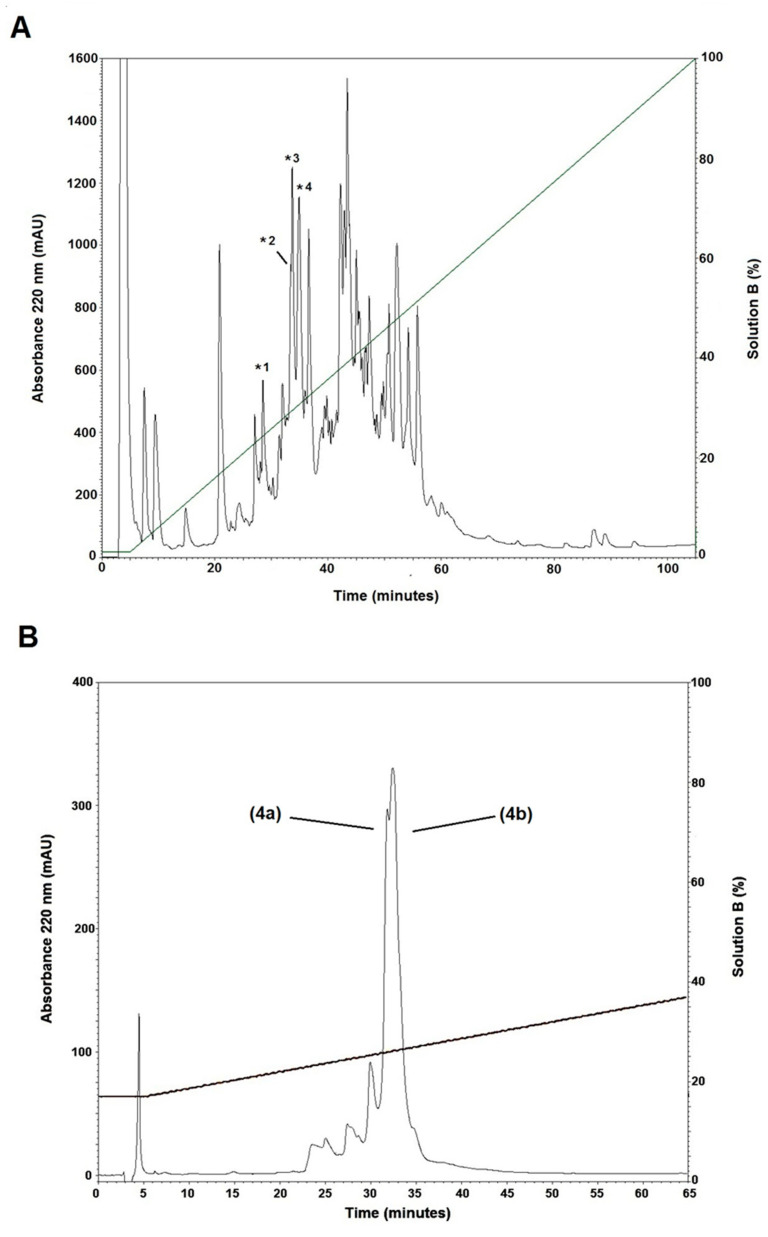
(**A**) Chromatogram profile of the fractionation of the crude venom from *C. fergusoni*. After 5 min at 5% Solution B, a linear gradient of 5 to 100% Solution B in 100 min was applied at a flow rate of 1 mL/min. Numbers *1–*4 indicate those fractions that presented activity over 90% on α7 nAChR. (**B**) The peak of the fourth active fraction, * 4 in A, was subfractionated using an isocratic step at 17% Solution B for 5 min, followed by an elution gradient of 17 to 37% Solution B in 60 min at a flow rate of 1 mL/min. This procedure yielded two major fractions, 4a and 4b.

**Figure 2 toxins-14-00510-f002:**
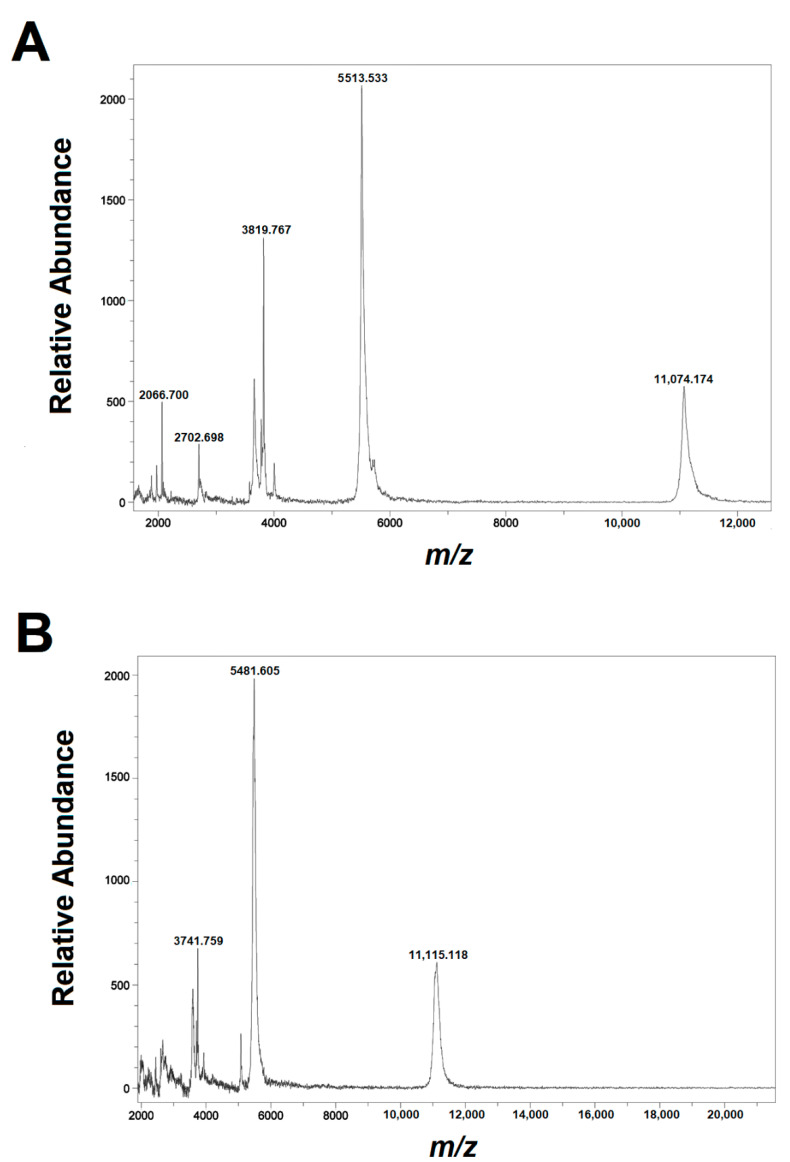
Mass spectra of the native semi-purified peptides obtained from the fourth active fraction from *C. fergusoni* venom. (**A**) Subfraction 4a; (**B**) Subfraction 4b.

**Figure 3 toxins-14-00510-f003:**
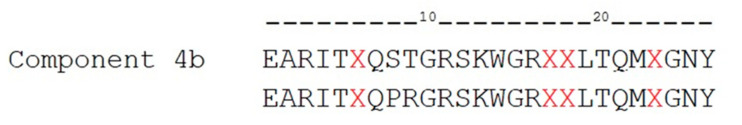
Partial amino sequence from component 4b by Edman degradation. X in red = Blank cycles that are assumed as cysteine residues.

**Figure 4 toxins-14-00510-f004:**
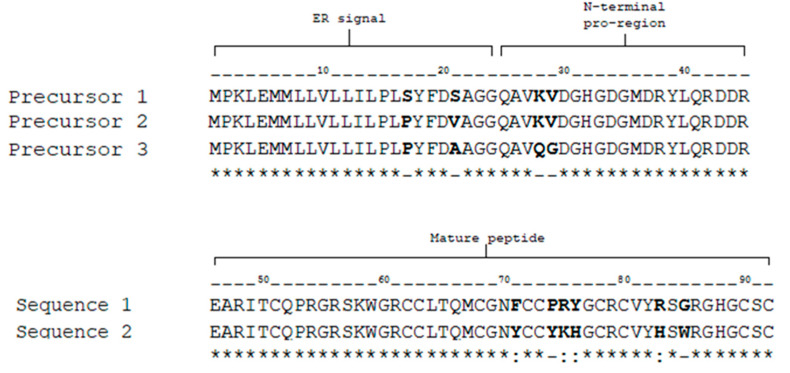
Transcriptomics of the αD-conotoxins present in *C. fergusoni* venom. * = same amino acids comparing both sequences; _ = different amino acids from a distint physicochemical group (bold face); : = different amino acids but from the same physicochemical group (bold face).

**Figure 5 toxins-14-00510-f005:**
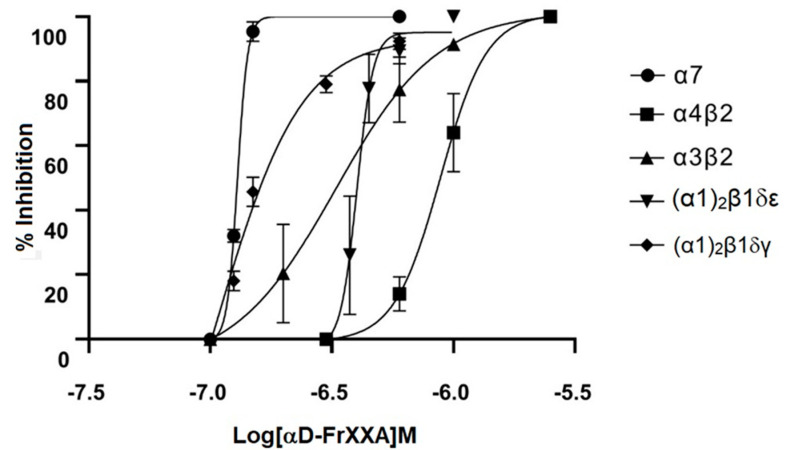
Dose-response curves of αD-FrXXA over different nAChR sybtypes. αD-FrXXA inhibits all subtypes with an IC_50_ for hα7 = 125 nM, hα4β2 = 697 nM, hα3β2 = 282 nM, m(α1)_2_β1δε 351 nM and m(α1)_2_β1δγ 447 nM.

**Figure 6 toxins-14-00510-f006:**
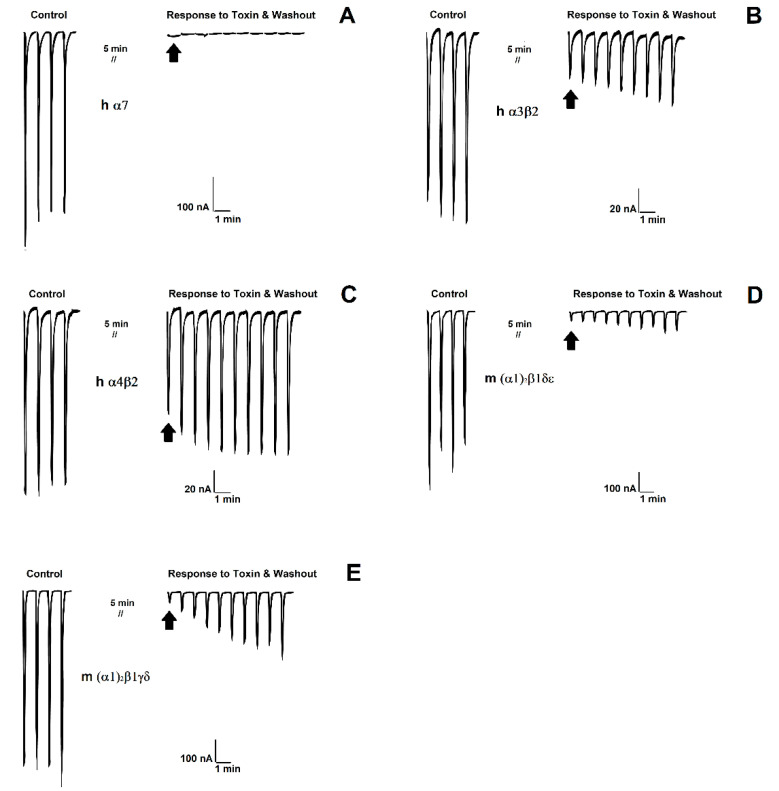
*Xenopus laevis* current recordings on five different subtypes (**A**) hα7, (**B**) hα3β2, (**C**) hα4β2, (**D**) m(α1)_2_β1δε, and (**E**) m(α1)_2_βδγ of nAChRs. At left, control currents elicited by 1 s pulses of 1mM acetylcholine prior to toxin addition. At right, peak current amplitudes at the end of five minutes incubation in 600 nM αD-FrXXA followed by washout of the toxin with agonist stimulation at one-minute intervals. ^//^ indicates that ND96 flow was stopped for 5 min.

**Table 1 toxins-14-00510-t001:** Comparison between experimental masses of compounds detected in Fractions 4a and 4b and theoretical masses of sequences determined by transcriptomic analysis, including probable chemical modifications introduced during the purification process.

Compound	tm/em	+ 1 TFA + 1 K	+ 2 TFA + 1 K + 1 O	+ 1 TFA + 1 K + 1 Na + 2 O	+ 2 TFA	+ 2 TFA + 1 O	em-tm
Dimer 2-2	10,872.60	11,023.70 ^a^		11,077.68 ^c^		11,114.62 ^e^	
Dimer 2-2 [P8S, R9T]	10,742.36		11,022.47 ^b^		10,968.38 ^d^		
4a-11,025.05 ^a, b^	11,025.05						+1.35 ^a^, +2.58 ^b^
4a-11,073.16 ^c^	11,073.16						−4.52 ^c^
4b-10,961.19 ^d^	10,961.19						−7.19 ^d^
4b-11,114.11 ^e^	11,114.11						−0.51 ^e^

All values are expressed as Da and are average masses. Some figures have been rounded to two digits compared to the values in the text. Letters in superscript (a–e) indicate correspondence between experimental and theoretical masses and their difference, both calculated taking into account the adducts and oxidations. Trifluoroacetic acid adducts, potassium adducts, and sodium adducts are denoted simply by TFA, K, and Na, respectively. O indicates oxidation, probably of M and/or W residues; tm, theoretical mass; em, experimental mass.

**Table 2 toxins-14-00510-t002:** IC_50_ of αD-conotoxins with pharmacological activity against nAChRs.

Conotoxin	nAChR	Species	IC_50_
GeXXA	α7	*H. sapiens*	210 nM
FrXXA	α7	*H. sapiens*	123 nM
PiXXA	α7	*H. sapiens*	6.2 nM
VxXXB	α7	*R. norvegicus*	400 pM
FrXXA	α4β2	*H. sapiens*	607 mM
GeXXA	α4β2	*R. norvegicus*	>3 nM
VxXXB	α4β2	*R. norvegicus*	228 nM
FrXXA	α3β2	*H. sapiens*	282 mM
GeXXA	α3β2	*R. norvegicus*	498 nM
VxXXA	α3β2	*R. norvegicus*	370 nM
VxXXB	α3β2	*R. norvegicus*	8.4 nM
GeXXA	α9α10	*H. sapiens*	28 nM
GeXXA	α9α10	*R. norvegicus*	1.2 nM
FrXXA	α1β1δε	*H. sapiens*	351 nM
GeXXA	α1β1δε	*R. norvegicus*	743 nM
VxXXB	α1β1γδ	*R. norvegicus*	3.5 nM
FrXXA	α1β1γδ	*H. sapiens*	447 nM

## Data Availability

Not applicable.

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
