# Peer review of "A Novel Dimeric Conotoxin, FrXXA, from the Vermivorous Cone Snail Conus fergusoni, of the Eastern Pacific, Inhibits Nicotinic Acetylcholine Receptors"

_toxins, 2022, doi:10.3390/toxins14080510_

Round 1

Reviewer 1 Report

It is a nice research work that isolated and characterized a novel alpha-D conotoxins, αD-FrXXA, from Eastern Pacific.  I have a few minor comments.

1. page 4, line90, 'major (z = +2)' could be modified to 'major m/z(z=+2) ' in order to be consistent with the following description.

2. page 12 line 320-21, "A value similar to the was obtained for αD-FrXXA (125 nM)". the sentence is incomplete.

3. page 9, Table 1. superscript letters (a-e) are confusing and need to be explained in more detail. 

Author Response

Reviewer 1

It is a nice research work that isolated and characterized a novel alpha-D conotoxins, αD-FrXXA, from Eastern Pacific.  I have a few minor comments.

  1. page 4, line90, 'major (z = +2)' could be modified to 'major m/z(z=+2) ' in order to be consistent with the following description.

ANSWER: Thank you very much for your observation, we have modified the text.

  1. page 12 line 320-21, "A value similar to the was obtained for αD-FrXXA (125 nM)". the sentence is incomplete.

ANSWER: Thank you again; we have rephrased this sentence in order to be clearer.

  1. page 9, Table 1. superscript letters (a-e) are confusing and need to be explained in more detail. 

ANSWER: Thank you very much for your remark. We have modified the footnote to avoid any confusion.

Reviewer 2 Report

This manuscript investigates a novel alpha conotoxin from Conus fergusoni.  Cone snails are the source of an incredible array of pharmacological agents, particularly for nicotinic receptors and voltage-gated ion channels.  A PubMed search for Conus fergusonii found no references (although the species is in Wikipedia), suggesting that the toxins from this species are unknown, and this may be the first report.  The authors provide evidence that the FrXXA conotoxin isolated from this snail is a member of the dimeric conotoxin family, a relatively new class of conotoxins first described in 2006.  This dimeric conotoxin shows almost irreversible blocking activity at human alpha7 nicotinic receptors and more reversible effects at a variety of other neuronal and muscle nicotinic receptors. This reviewer is not qualified to comment on the quality of the mass spectroscopy portion of this manuscript (hopefully the other reviewers are), but the other biochemical isolation and characterization of the toxin is given in enough detail that should allow others to replicate these findings.  This manuscript should be of general interest to pharmacologists interested in nicotinic receptors in addition to specialists in conotoxins.

Having said that, it’s clear that this manuscript could use some revisions to fix minor English issues.  Some examples:

In line 157, there are multiple dose response curves, so “curve” should be plural “curves”.  Also “FrXXA inhibit” should be “FrXXA inhibits”.

Also, on lines 163-164, not sure what “At left, peak currents amplitude before toxin proved.” means. Perhaps it would be better if the sentences read “At left, control currents elicited by 1 second pulses of 1 mM acetylcholine prior to toxin addition.  At right, peak current amplitudes at the end of five minutes incubation in 600 nM alphaD-FrXXA followed by washout of the toxin with agonist stimulation at one-minute intervals.”

Also, the paragraph lines 274-280, not sure what the authors are saying here.  Are they saying that dimer formation has only been experimentally demonstrated for conotoxin peptides from C. vexillium and C. generalis? Also, there should be no comma after “least” in line 276.

In line 307, “mean” should be “means” and what follows is not clear.  Are they saying that unlike other alpha conotoxins, the Dalpha- and psi-conotoxins are non-competitive antagonists that bind to sites distinct from orthosteric agonist binding sites?

What is the union referred to in line 317?  Are they referring to a dimer?

These are just some examples.

Also, it would have been nice if the authors had tested the alpha9/alpha10 subtype since other dimeric conotoxins inhibit those as well. Another wish-list item would be to get the X-ray crystal structure, but that would require purifying or synthesizing much more toxin.

Author Response

This manuscript investigates a novel alpha conotoxin from Conus fergusoni.  Cone snails are the source of an incredible array of pharmacological agents, particularly for nicotinic receptors and voltage-gated ion channels.  A PubMed search for Conus fergusonii found no references (although the species is in Wikipedia), suggesting that the toxins from this species are unknown, and this may be the first report.  The authors provide evidence that the FrXXA conotoxin isolated from this snail is a member of the dimeric conotoxin family, a relatively new class of conotoxins first described in 2006.  This dimeric conotoxin shows almost irreversible blocking activity at human alpha7 nicotinic receptors and more reversible effects at a variety of other neuronal and muscle nicotinic receptors. This reviewer is not qualified to comment on the quality of the mass spectroscopy portion of this manuscript (hopefully the other reviewers are), but the other biochemical isolation and characterization of the toxin is given in enough detail that should allow others to replicate these findings.  This manuscript should be of general interest to pharmacologists interested in nicotinic receptors in addition to specialists in conotoxins.

Having said that, it’s clear that this manuscript could use some revisions to fix minor English issues.  Some examples:

In line 157, there are multiple dose response curves, so “curve” should be plural “curves”.  Also “FrXXA inhibit” should be “FrXXA inhibits”.

ANSWER: Thank you very much, we have done the suggested change.

Also, on lines 163-164, not sure what “At left, peak currents amplitude before toxin proved.” means. Perhaps it would be better if the sentences read “At left, control currents elicited by 1 second pulses of 1 mM acetylcholine prior to toxin addition.  At right, peak current amplitudes at the end of five minutes incubation in 600 nM alphaD-FrXXA followed by washout of the toxin with agonist stimulation at one-minute intervals.”

ANSWER: Thank you very much again, we followed your recommendation.

Also, the paragraph lines 274-280, not sure what the authors are saying here.  Are they saying that dimer formation has only been experimentally demonstrated for conotoxin peptides from C. vexillium and C. generalis? Also, there should be no comma after “least” in line 276.

ANSWER: Thank you for your remark; we added two more lacking species, C. capitaneus and C. princeps and the corresponding references. The comma was deleted.

In line 307, “mean” should be “means” and what follows is not clear.  Are they saying that unlike other alpha conotoxins, the Dalpha- and psi-conotoxins are non-competitive antagonists that bind to sites distinct from orthosteric agonist binding sites?

ANSWER: Thank you again; we have rephrased this sentence in order to be clearer.

What is the union referred to in line 317?  Are they referring to a dimer?

ANSWER: Thank you; we are not referring to a dimer, we referred to the binding site for families alphaD and Psi-conotoxins.

These are just some examples.

Also, it would have been nice if the authors had tested the alpha9/alpha10 subtype since other dimeric conotoxins inhibit those as well. Another wish-list item would be to get the X-ray crystal structure, but that would require purifying or synthesizing much more toxin.

ANSWER: Indeed, we share the wish-list! We tried to express the alpha9/alpha10 subtype of nAChR. However, this particular subtype did not express satisfactorily in our electrophysiology model. At least for now, the purification of enough amount of the native dimer is not possible, and the same applies for synthetic dimer.